# Protein-Based Adjuvants for Vaccines as Immunomodulators of the Innate and Adaptive Immune Response: Current Knowledge, Challenges, and Future Opportunities

**DOI:** 10.3390/pharmaceutics14081671

**Published:** 2022-08-11

**Authors:** Diego A. Díaz-Dinamarca, Michelle L. Salazar, Byron N. Castillo, Augusto Manubens, Abel E. Vasquez, Fabián Salazar, María Inés Becker

**Affiliations:** 1Fundación Ciencia y Tecnología para el Desarrollo (FUCITED), Santiago 7750000, Chile; 2Sección de Biotecnología, Departamento Agencia Nacional de Dispositivos Médicos, Innovación y Desarrollo, Instituto de Salud Pública de Chile, Santiago 7750000, Chile; 3Biosonda Corporation, Santiago 7750000, Chile; 4Facultad de Ciencias para el Cuidado de la Salud, Universidad San Sebastián, Providencia, Santiago 8320000, Chile; 5Medical Research Council Centre for Medical Mycology, University of Exeter, Exeter EX4 4QD, UK

**Keywords:** proteins, adjuvants, vaccines, protein-based adjuvants, Toll-like receptors, C-type lectin receptors, innate immune response, adaptive immune response

## Abstract

New-generation vaccines, formulated with subunits or nucleic acids, are less immunogenic than classical vaccines formulated with live-attenuated or inactivated pathogens. This difference has led to an intensified search for additional potent vaccine adjuvants that meet safety and efficacy criteria and confer long-term protection. This review provides an overview of protein-based adjuvants (PBAs) obtained from different organisms, including bacteria, mollusks, plants, and humans. Notably, despite structural differences, all PBAs show significant immunostimulatory properties, eliciting B-cell- and T-cell-mediated immune responses to administered antigens, providing advantages over many currently adopted adjuvant approaches. Furthermore, PBAs are natural biocompatible and biodegradable substances that induce minimal reactogenicity and toxicity and interact with innate immune receptors, enhancing their endocytosis and modulating subsequent adaptive immune responses. We propose that PBAs can contribute to the development of vaccines against complex pathogens, including intracellular pathogens such as *Mycobacterium tuberculosis*, those with complex life cycles such as *Plasmodium falciparum*, those that induce host immune dysfunction such as HIV, those that target immunocompromised individuals such as fungi, those with a latent disease phase such as *Herpes,* those that are antigenically variable such as SARS-CoV-2 and those that undergo continuous evolution, to reduce the likelihood of outbreaks.

## 1. Introduction

Vaccines have substantially reduced the burden of infectious diseases. In particular, the eradication of smallpox in 1980 through vaccination is one of the most significant medical achievements to date. However, many infectious diseases with worldwide significance are not currently preventable through vaccination. Traditional live-attenuated/inactivated whole-pathogen vaccines alone are sufficient to induce robust long-lasting immunity in mammals. However, these vaccines are unsuitable when natural infection does not confer long-term protection or when the pathogen cannot be grown in culture. Recombinant protein vaccines (composed of subunits or purified antigens) in isolation elicit only weak and short-lived immune responses [1]. Therefore, these vaccines must be delivered with an adjuvant to enhance and target the adaptive immune response to the vaccine antigens.

A vaccine-adjuvant is a substance or combination of substances added to a vaccine that enhance immunogenicity and contribute to an initial innate immune response by inducing an inflammatory reaction at the injection site. Thus, an adjuvant enhances the magnitude and durability of the vaccine and alters the effect of specific adaptive downstream immune responses to vaccine antigens without inducing a specific antigenic effect against itself [2,3,4,5,6]. In addition, the use of adjuvants benefits vaccine product development in several ways, including reduced antigen dosing and fewer necessary immunizations, which can exert a potential effect on the global vaccine supply. Adjuvants can be classified into different category types, and herein, we classified adjuvants on their the basis of their physicochemical properties into five families: chemicals (e.g., aluminum salts) [7], lipids (e.g., monophosphoryl lipid A [MPLA]) [8], polysaccharides (e.g., lipopolysaccharide [LPS]) [9], oligonucleotides (e.g., the CpG oligonucleotide) [10], and proteins (e.g., flagellin) [11]. In this review, we focus on protein-based adjuvants (PBAs) isolated from different organisms that show significant immunostimulatory properties.

The development of new vaccine adjuvants has been considered one of the slowest processes in the history of medicine. Since the initial licensure of aluminum salts in the 1920s, it has remained for more than 70 years as the only adjuvant included in licensed vaccines against hepatitis B, diphtheria, tetanus, and pertussis, among others [6,7,12]. The main difficulties encountered in developing new adjuvants include limited understanding of their molecular complexity, including the molecular mechanisms of PBA action that induce an immune response. The PBAs can exhibit one or more of the following properties: (i) binding and activation of innate immune receptors, known as pattern recognition receptors (PRR), such as Toll like receptors (TLRs), C-type lectin receptors (CLRs), NOD-like receptors, and RIG-I-like receptors (retinoic acid-inducible gene-I-like receptors, RLRs) which recognized pathogen-associated molecular patterns (PAMPs) [1,6,11]; (ii) induction of proinflammatory cytokines such as IL-1β, IL-6, TNF-α, and IL-12 and chemokines such as IL-8, which recruit innate immune cells [13]; (iii) regulated expression of co-signaling molecules required for T-cell activation, such as the B7 family members, on professional antigen-presenting cells (APCs) [14]; (iv) induction of specific humoral-mediated (B cell) or cell-mediated (T-cell) immune responses [15]; and (v) induction of trained innate immunity, harnessing the activation state of APCs to enhance adaptive T-cell responses to both the specific antigen and PBA, generating a beneficial bystander effect [16]. A better understanding of these mechanisms will pave the way for the development of next-generation PBA that can stimulate and increase the magnitude and durability of the adaptive immune response to vaccine antigens [6].

The innate and adaptive immune systems constantly interact with one another to generate an efficient immune response. In the early phases of an immune response, the innate immune system senses microbes through the PRRs expressed on APCs, influencing the subsequent adaptive immune response [17]. Thus, after the activation of these cells, the antigen processing progresses [18]. Subsequently, protein antigens in the form of peptide fragments are carried to the surface of APCs on Major Histocompatibility Complex class I (MHC-I) and class II (MHC-II) proteins, stimulating antigen-specific T-cell responses, and thus develop the cellular-mediated adaptive immunity, which involves the activation of antigen-specific helper T-cells and cytotoxic T-cells [19]. In T-cell-dependent B-cell responses, both respond to the same antigen, and B-cells serve as antigen presenting cells that capture their cognate antigen by surface immunoglobulin. This antigen is internalized, processed, and presented to T-helper cells, increasing MHC-II expression and the reciprocal expression of costimulatory molecules (CD80, CD86, CD40) [20]. This process is promoted by the cytokines secreted by T-helper cells. As a result, B-cells mature and produce the antibody- or humoral-mediated adaptive immunity [18,21].

This review focuses on PBAs not only for the formulation of vaccines to prevent infectious disease but also in vaccines for treating existing cancer, known as therapeutic cancer vaccines. We provide information on the current knowledge of certain highly relevant PBAs that show great promise as adjuvants. The main PBA structural features and proposed mechanisms of PBA action are presented. We also highlight basic preclinical studies and clinical research phase studies for PBA evaluation and current challenges to vaccine PBA development, covering recombinant PBAs, pharmaceutical and regulatory considerations, and clinical safety concerns. Finally, we include a section on potential opportunities for PBA use in new-generation human vaccines. We selected PBAs that meet one or more of the aforementioned criteria. The emphasis is placed on classical and encouraging PBAs derived from bacteria, marine mollusks, plants, and humans.

## 2. PBAs as Agonists of Innate Immune Receptors

Since 1989, when Charles Janeway proposed the pivotal theory explaining the mechanism by which the innate immune system discriminates between autoantigens and pathogens, pattern recognition receptors (PRRs) have received a significant attention. Different PRRs families expressed on APCs recognize pathogen-associated molecular patterns (PAMPs) and damage-associated molecular patterns (DAMPs), leading to their activation and subsequent influencing the type of adaptive immune response [22,23]. The TLR family has received particular attention. Notably, TLR1, TLR2, TLR4, TLR5, and TLR6 are expressed on the cell surface of APCs, whereas TLR3, TLR7, TLR8, and TLR9 are expressed in endosomes [24]. Through their ectodomains, TLRs bind a wide variety of pathogenic substances [25].

The TLR ligand induces the dimerization of TLRs and recruitment of adaptor proteins to intracellular Toll/interleukin-1 receptor (TIR) domains to initiate signaling [17,26]. The signaling cascades triggered via these TIR domains are mediated by specific adaptor molecules, including myeloid differentiation primary response 88 (MyD88); MyD88-adaptor-like (MAL), also known as TIR domain-containing adaptor protein (TIRAP); TIR domain-containing adapter-inducing IFNβ (TRIF); and TRIF-related adaptor molecule (TRAM) [26]. These adaptor proteins have TIR domains and establish TIR–TIR interactions that can be categorized as receptor–receptor, receptor–adaptor, and adaptor–adaptor interactions. Combinatorial recruitment of these adaptors via TIR–TIR interactions orchestrates downstream signaling, leading to the induction of proinflammatory gene expression [27,28] (Figure 1).

The Tollike receptors TLR1 and TLR6 heterodimerize with TLR2 and engage in signaling through the MyD88 pathway to activate NF-κB and MAP kinases, which leads to cytokine secretion [6,17,29]. The receptors TLR4 and TLR5 are homodimers in the MyD88 signaling pathway. In addition, TLR4 engages with TRIF to induce type I interferon expression mediated through IRF [6,17,29]. Because of the roles they play in controlling APC-immunomodulatory functions, TLR agonists are considered promising vaccine adjuvants candidates. Most TLR-binding PBAs activate TLR2, TLR4, and TLR5 to mediate the innate immune response. In addition, other PBAs bind to C-type lectin receptors and the ganglioside GM1/GD1a to trigger immune effects, which are described in the following subsections.

The PBAs can interact at with TLR2 (e.g., porins FomA from *Fusobacterium nucleatum* and major outer membrane protein [MOMP] from *Shigella dysenteriae*, AB-type toxins from *Vibrio cholerae,* lipoprotein OMP 16 from *Bordetella pertussis*); with TLR4 (e.g., high-mobility group box 1 [HMGB1], heat-shock-70-like protein 1 from humans, surface immunogenic protein [SIP] from Group B *Streptococcus* (GBS), pneumolysin from *Streptococcus pneumoniae*, hemocyanin from *Concholepas concholepas* and mistletoe lectin-I [ML-I] from *Viscum album*); with TLR5 (e.g., flagellin); and heterodimers TLR2–TLR1 (e.g., porin OmpU from *V. cholerae*) or TLR2–TLR6 (e.g., porin from *S. dysenteriae* (Figure 1).

### 2.1. TLR-2-Dependent Activation by PBAs

Because of its ability to recognize ligands as a heterodimer associated with TLR1 or TLR6, TLR2 detects many PAMPs from a wide variety of pathogens [30], as summarize in Table 1. In addition, many reports have indicated that, after heterodimerization, TLR2 signaling triggers a pro- or anti-inflammatory response. The TLR2 interactions with PAMPs may lead to CD4+ T-helper lymphocyte (Th1 and Th2 cell) or regulatory T (Treg) cell differentiation [6].

Among PBA agonists implicated in the activation of TLR2 are bacterial pore-forming proteins known as porins [11]. Porins are outer-membrane proteins (OMPs) that form channels in Gram-negative bacteria, which regulates the entry of ions, such as K+ and Cl-, and small substrates [31]. The monomeric porin structure is approximately 48 kDa [32]. Porins are highly conserved among bacteria. All porins form homotrimers, although some dimeric or octameric porins have been described; these structural variations correlate with differential permeabilities of the OMP-formed pores [33,34,35].

Porins from *Fusobacterium nucleatum* (FomA), a human oral pathogen [36], induces IL-6 secretion and cell surface upregulation of CD86 and major histocompatibility complex (MHC) type II in splenic B cells. A recombinant form of FomA has been obtained, and its immunostimulatory properties, which are mediated through TLR2 signaling in vitro and in vivo, have been evaluated. Recombinant FomA induces a Th2-cell-type adjuvant effect characterized by enhanced production of OVA-specific IgG1 and IgG2b antibodies in C57BL/6 mice and enhanced secretion of IL-10 and IL-6 by splenic B cells [37].

The major OMP (MOMP) from *Shigella flexneri* activates TLR2, enhancing NF-kB and p38 MAP kinase activation [38]. In addition, the MOMP from *Chlamydia trachomatis* has been shown to induce IL-8 and IL-6 production in a TLR2/TLR1-dependent manner [39].


pharmaceutics-14-01671-t001_Table 1Table 1Bacterial PBAs with TLR2 adjuvant potential.SpeciePBAInnate Immune ResponseAdaptive Immune ResponseReference
**
*Fusobacterium*
**
FomA PorinProinflammatory cytokines and costimulatory cytokines OVA-Specific antibodies in a preclinical model[36,37]
**
*Shigella flexneri*
**
Major outer membrane protein (MOMP) Proinflammatory profileND[38]
**
*Shigella dysenteriae*
**
PorinActivation of mitogen-activated protein kinase (MAPK) and nuclear factor B (NF-B). Up-regulation of CD80; MHC class II; and CD40. B-cell activation.Th1 immune response[41,42,43]
**
*Vibrio cholerae*
**
OmpUM1 polarization and NF-κB activationND[44]
**
*Neisseria meningitidis*
**
PorBPro-inflammatory cytokines, activation of APCsIncreases of Follicular Dendritic cells; Th1 immune response; Antigen-Cross presentation[45,46]
**
*Salmonella typhi*
**
OmpC and OmpFPro-inflammatory cytokines, activation of APCsIncrease in IgG antibody titers[47]
**
*Mycobacterium tuberculosis*
**
Early secreted antigenic target protein 6 (ESAT-6)Proinflammatory cytokines Th17 immune response with a role in protection against *M. tuberculosis*infection[48]
**
*Brucella abortus*
**
rBCSP31Proinflammatory cytokines; priming of CD4+ T-cellsTh1 type immune response protects against *B. abortus*Infection.[49]
**
*Streptococcus pneumoniae*
**
DnaJ-ΔA146Plyproduction of IL-12 in BM-DCsTh1 Immune Response against *Streptococcus pneumoniae*[50]
**
*Streptococcus pneumoniae*
**
Endopeptidase O (PepO)Proinflammatory cytokines ND[51]ND: Not determined.


Porin from *Shigella dysenteriae* stimulated TLR2/TLR6 naïve CD4+ T-cells, B-cells, and macrophages [41,42,43], contributing to a Th1-cell immune response. In addition, the *Vibrio cholerae* porin OmpU mediates M1-polarization of macrophages/monocytes via TLR1/TLR2 activation [44]. The PorB porin from *Neisseria meningitidis* preferentially binds to the TLR2/TLR1 heterodimer compared to the TLR2/TLR6 heterodimer, upregulating CD86 expression in splenic B cells and NF-κB nuclear translocation in a HEK reporter cell line [45]. The PorB porin also enhances APC trafficking and cross-presentation and increases antigen deposition on germinal center follicular dendritic cells (DCs) [46].

Immunization against an inactivated H1N1 2009 pandemic influenza virus combined with *Salmonella enterica* serovar *Typhi* porins OmpC and OmpF as adjuvants, elicits a humoral response characterized by higher hemagglutinating anti-influenza IgG titers, antibody class switching rates, and affinity maturation. In addition, coadministration of OmpC and OmpF with unconjugated Vi capsular polysaccharide (a T-cell-independent antigen) induces higher IgG antibody titers and class switching rate in a murine model. The mechanism mediating these adjuvant effects might be related to the agonistic effect of *S. typhi* porins on TLR2 and TLR4 activity [47].

Tuberculosis (TB), caused by *Mycobacterium tuberculosis,* remains a major infectious disease worldwide. Early secreted antigenic target protein 6 (ESAT-6) is one of the most prominent antigens expressed by *M. tuberculosis* strain H37Rv. This PBA promotes a lung Th17 immune cell response in a TLR2-dependent manner. In addition, ESAT-6 induces IL-6 and TGF-β production by DCs [48].

Moreover, recombinant *Brucella* cell-surface protein 31 (rBCSP31) from *Brucella abortus* is a TLR2 and TLR4 protein agonist that induces TNF-α, IL-6, and IL-12p40 cytokine production by macrophages and a Th1-cell immune response [49].

*Streptococcus pneumoniae*, one of the leading causes of invasive bacterial disease worldwide, expresses two heat shock proteins that are important conserved virulence factors: DnaJ and pneumolysin (Ply), which are TLR4 and TLR2 ligands. The protein DnaJ is a member of the Hsp40 family and functions mainly as a molecular Hsp70 chaperone and thus participates in protein folding and assembly. Notably, Ply is a toxin that can be used as a carrier protein with future pneumococcal conjugate vaccines because of its immunogenic activity; however, it is highly toxic [52], which makes incorporating Ply into new vaccines a challenge.

A fusion protein comprising two virulence factors of *Streptococcus pneumoniae*, DnaJ and a less-toxic Ply mutant (DnaJ-ΔA146Ply), induces the production of IL-12 and Th1 cell proliferation mediated via TLR2 in bone marrow-derived dendritic cells (BMDCs). In addition, in a preclinical model, DnaJ-ΔA146Ply confers protection against *S. pneumoniae* in a TLR2-dependent manner [50]. Another protein agonist of TLR2 in *S. pneumoniae* is recombinant endopeptidase O (rPepO), a pneumococcal virulence protein. The intratracheal instillation of rPepO protein results in a significant increase in IL-6, TNF-α, CXCL1, and CXCL10 production and neutrophil infiltration in mouse lungs. Interestingly, compared with wild-type mice, TLR2- or TLR4-deficient mice subjected to rPepO treatment show decreased cytokine production, reduced neutrophil infiltration, and intensified tissue injury. In addition, upon stimulation of peritoneal exudate macrophages (PEMs), rPepO induces IL-6, TNF-α, CXCL1, and CXCL10 production, which relies on the rapid phosphorylation of p38, protein kinase B (PKB, also known as Akt), and p65 in a TLR2-/TLR4-dependent manner [51].

Lipoproteins, such as BP1569 from *B. pertussis*, have emerged as novel TLR2 agonists. BP1569 has a molecular mass of 40 kDa and shares a sequence with lipoproteins from *N. meningitidis*, *Burkholderia pseudomallei*, and *Haemophilis influenzae* [53,54]. The three-dimensional structure of BP1569 and that of other lipoproteins of interest have not been characterized; however, comparative analyses of the amino acid sequences indicates that these proteins contain a positively charged N-terminal signal sequence, followed by a hydrophobic region and a lipobox sequence, which is acylated. Notably, the lipobox acyl group is essential for the immunostimulant effects of these lipoproteins since it directly interacts with TLR2 [55].

### 2.2. TLR2-and Ganglioside-Dependent Activation by PBAs

AB-type toxins, such as cholera toxin (CT) in *V. cholerae* and heat-labile enterotoxin (LT) in *Escherichia coli*, have been extensively studied as mucosal adjuvants. Table 2 resumes PBAs of bacterial origin with TLR2 and GM1ganglioside adjuvant potential. The CT and LT show high amino acid sequence identity, and their three-dimensional structures are similar [56,57,58]. Cholera toxin is a hexamer formed by a single A subunit (28 kDa) and five B subunits (11 kDa each). The A subunit comprises A1 and A2 domains: A1 is a globular ADP-ribosylase whereas A2 is an extended alpha-helix. The A2 domain tethers the A subunit to the pentameric ring formed by the B subunits [58]. The B5 pentameric ring is essential to pathogenesis because it binds to glycosphingolipids on target cells, such as the ganglioside GM1, allowing CT endocytosis. This process promotes the release of A1 after disulfide bridges reduction and the subsequent recognition, processing, refolding, and activation of A1 enzymatic activity [59,60].

Similarly, the LT structure is an AB5 hexamer: a single A subunit with catalytic activity, and five B subunits form a ring with membrane-binding functions [57]. However, CT and LT present biophysical differences in terms of solvent-accessible contact area: CT has a higher contact area than LT, which limits the diffusion of water through CT-formed pores. These differences correlate with the different pathogenic effects of CT and LT [61]. Full-length CT and LT are pathogenic, and therefore, mucosal adjuvants are composed of less-toxic and less-allergenic derivatives of these proteins [62]. In one strategy, the catalytic activity of CT and LT is prevented by administering only the pentameric ring, which has immunomodulatory properties itself [63]. Another strategy involves site-directed mutagenesis to impair the catalytic activity of the A subunit. These mutants are powerful immunostimulants, but they might show residual activity depending on the amino acid residue substitution [64].

Cholera toxin enters the endoplasmic reticulum of immune cells through endosomes following binding to the ganglioside GM1 in mucosal membranes [65]. Then, the A1 subunit is released via disulfide bond reduction, and CTA1 is retroactively translocated to the cytosol [66]. Cytosolic CTA1 can bind to Gsα, catalyzing its ADP-ribosylation, and subsequently can elevate the 3′,5′-cyclic AMP (cAMP) concentration in a host cell [67]. Enhanced cAMP concentration mediated through cytosolic CTA1 induces the production of proinflammatory cytokines, including IL-1β, TNF-α, and IL-6, in DCs and CD4+ T-cells [68]. In a model of intranasal anthrax infection, CT has been shown to mediate the induction of IL-17-producing CD4+ Th17 cells [68] (Figure 2).

*Escherichia coli* type I and type II LTs share many physiological and structural features. However, recent studies have shown that each toxin triggers unique signaling cascades, leading to different cellular responses [69]. The LTs interact with gangliosides, mediating the signaling between immune-competent cells, in which the composition of ganglioside species varies. However, LT B-pentamers typically interact with either gangliosides and/or TLRs. Studies have established that the B-pentamers in type II LTs (LT-IIa and LT-IIb) interact with TLR2, leading to the induction of IL-1β, IL-6, IL-8, and TNF-α expression in human THP-1 cells [69]. Furthermore, stimulation of human embryonic kidney (HEK)-293 cells that transiently express TLR1 and TLR2 has been shown to activate NF-κB-dependent luciferase gene expression [69,70]. In addition, studies have demonstrated the importance of the interaction between TLR2 and GD1a and a subunit of type IIb *E. coli* enterotoxin (LT-IIb-B5) [71]. Both LT-IIb-B5 and a defective GD1a-binding mutant (LT-IIb-B5[T13I]) binds TLR2 with moderate affinity. However, only the wild-type molecule demonstrates a significant increase in TLR2-binding activity in the presence of GD1a. Furthermore, fluorescence resonance energy transfer experiments have indicated that LT-IIb-B5 induces the recruitment of TLR2 and TLR1 to lipid rafts and clustering with GD1a, in contrast to the defective GD1a-binding mutant, which does not activate TLR2 signaling [72].


pharmaceutics-14-01671-t002_Table 2Table 2Bacterial PBAs with TLR2 and GM1 Ganglioside adjuvant potential.SpeciePBAInnate Immune ResponseAdaptive Immune ResponseReference
**
*Vibrio cholerae*
**
Cholera toxinPro-inflammatory cytokines in DCs and CD4 T-cells.Induction of IL-17-producing CD4+ Th17 cells[67,68]
**
*Escherichia coli*
**
B-pentamers of the type II LTs LT-IIa and LT-IIbPro-inflammatory cytokines in humans THP-1 cells and NF-κBND[69,70,71,72]ND: No determined.


### 2.3. TLR4-Dependent Activation by PBAs

The receptor TLR4 is a promising target for immunomodulation, partially due to the success of the GlaxoSmithKline-produced MPLA adjuvant, the first TLR agonist approved by the Food and Drug Administration (FDA) for use in the development of new vaccines. Moreover, the adjuvant AS01, a mixture of MPL and QS21 (purified saponin from *Quillaja saponaria*), activates TLR4, stimulating a Th1-cell-type response, which can trigger the activation of CD8+ T-lymphocytes, showing potential applications to vaccines for malaria and herpes zoster [73]. There are several TLR4-dependent PBAs that exhibit great immunogenicity amongst them cell death derived adjuvants (HMGB1 and Hsp70L1), SIP from GBS, virulence factors DnaJ and Ply from *S. pneumoniae*, Omp16 from *Brucella abortus*, proteins from *M. tuberculosis* (RpfE, HBHA, Rv0652 and GrpE), hemocyanins and plant lectins. Table 3 resumes PBAs from different organism with TLR4 adjuvant potential.

It has been described that adjuvants such as Alum or MF59 cause local tissue damage and cell death creating a local pro-inflammatory milieu to recruit immune cells [3,74,75,76]. In this context, the extracellular release of HMGB1 can activate DCs to stimulate adaptive immunity [77]. Another mammalian protein regulating cell death is Hsp70-like protein 1 (Hsp70L1), promotes the production of TNF-α, IL-1, and IL-12p70 and the expression of surface markers such as CD40, CD80, and CD86 in bone marrow-derived DCs [78]. On the other hand, in a murine model, Hsp70L1 generates a specific Th1-cell-triggered immune response against carcinoembryonic antigen (CEA) [78].

The Surface Immunogenic Protein (SIP) in in Group *B Streptococcus* (GBS) has a molecular mass of 53 kDa, and its amino acid sequence is highly conserved among different species. The main secondary SIP structure consists of β-sheets, but the three-dimensional structure has not been characterized [79]. Similarly, the physiological function of SIP remains unknown; however, this protein is exposed to GBS surface and can be secreted [80,81]. Furthermore, a recombinant surface immunogenic protein in GBS (rSIP), expressed by *E. coli* and *Pichia pastoris* has shown immunomodulatory properties as a TLR4 agonist protein adjuvant [79,82]. Our data show that rSIP stimulates innate immune cells as an adjuvant to induce Th1 adaptive immune responses and is an oral mucosal vaccine candidate against GBS [82,83,84,85] (Figure 3).

Virulence factors of *S. pneumoniae*, such as DnaJ [86] and Ply [51,87], have been shown to act as TLR4 ligands. In this context, pneumolysin stimulates TNF-α and IL-6 in wild-type macrophages but not in macrophages in which MyD88 is deleted. Moreover, macrophages that carry a spontaneous mutation in TLR4 (P712H) are hyporesponsive to Ply. Recombinant DnaJ induces BMDC activation and maturation mediated via TLR4 and activated MAP kinase, NF-κB, and PI3K-Akt pathways. In addition, rDnaJ-treated BMDCs effectively stimulated naïve CD4+ T-cells to secrete IFN-γ and IL-17A. Moreover, the fusion of DnaJ and a less toxic Ply mutant (1A146Ply-) DnaJ-1A146Ply induces TLR4-dependent Th1- and TH17-cell-like responses against *S. pneumoniae* [88].

The outer membrane protein (Omp)16 lipoprotein is another TLR4 ligand, from *Brucella abortus*, that stimulates DCs and macrophages in vitro and induces a protective TLR4-dependent Th1-cell immune response against *B. abortus* infection [89]. Wild-type macrophages and BMDCs pulsed with Omp16 showed a significant increase in IL-12 and TNF-α expression compared to cells from TLR4-deficient mice. In addition, immunization with Omp16 generates Th1-cellimmune responses characterized by the secretion of IFN-γ by murine splenocytes [89].


pharmaceutics-14-01671-t003_Table 3Table 3PBAs from different organisms with TLR4 adjuvant potential.SpeciePBAInnate Immune ResponseAdaptive Immune ResponseReference
**
*Brucella abortus*
**
Omp16Induction of proinflammatory cytokines, and APC activationTh1 immune response against *B. abortus* infection[89]
**
*Brucella*
**
** spp.**
BLSPro-inflammatory cytokinesND[90]
**
*Mycobacterium tuberculosis*
**
GrpE; RpfE; Rv0652; HBHAInduction of proinflammatory cytokines, and DCs activationTh1 immune response[91,92,93,94]
**
*Streptococcus*
**
** Group B (GBS)**
Recombinant Surface Immunogenic Protein (rSIP)Pro-inflammatory cytokinesIncrease of humoral immune response and protection against GBS[79,82,83,85]
**
*Streptococcus pneumoniae*
**
Pneumolisin(NF-κB) activation and secretion TNF-α; IL-6)ND[49,87]
**
*Streptococcus pneumoniae*
**
DnaJMAPKs, NF-B and PI3K-Akt activationTh1 and Th17 activation[86,88]
**
*Mycobacterium avium*
**
** subsp**
MAP CobTMAP kinases and NF-kB activationTh1 immune response[95]
**Human**
HSP70L1NF-κB and MAPKs activation. Secretion of proinflammatorycytokinesTh1 immune response[78]
**Human**
high mobility group box 1 proteins (HMGB1)(NF-κB) activation and secretion of proinflammatorycytokines (TNF-α; IL-6; IL-1β)ND[96,97]
**Mollusk Hemocyanins**
FLH; CCH; KLHPro-inflammatory cytokines and ERK1/2 phosphorylationTh1 immune response[98,99,100,101,102,103,104,105]
**Plant Lectins**
Mistletoe lectin I; Soybean agglutinin; *Mistletoe lectin;* JacalinNF-κB activationIncrease humoral immune response and Th2 immune response[106,107,108,109,110,111]ND: Not determined.


*Mycobacterium tuberculosis* harbors four TLR4-dependent protein ligands including RpfE, HBHA, Rv0652 and Grp. Resuscitation-promoting factor E (RpfE), a latency-associated member of the Rpf family, promotes naïve CD4+ T-cell differentiation toward Th1 and Th17 cells. The RpfE induces DC maturation by increasing the expression of surface molecules such as CD86 and CD80 and the production of IL-6, IL-1β, IL-23p19, IL-12p70, and TNF-α [91]. In this context, RpfE activates ERK, p38 MAPKs, and NF-κB signaling after TLR4 binding [91].

Heparin-binding hemagglutinin adhesin (HBHA) induces DC maturation in a TLR4-dependent manner that is characterized by the expression of CCR7, CD40, CD80, CD86, MHC class I and II, and the proinflammatory cytokines IL-6, IL-12, IL-1β, and TNF, leading to a Th1-clell immune response. In addition, mechanistic investigations have established that the MyD88 and TRIF signaling pathways downstream of TLR4 mediate the secretion of HBHA-induced proinflammatory cytokines [92]. The 50S ribosomal protein L7/L12 (RPLL) Rv0652 modulates DC maturation and proinflammatory cytokine production (TNF-α, IL-1β, and IL-6) partially mediated through the TLR4/MyD88 signaling pathway. Moreover, DCs pulsed with Rv0652 plus OVA exhibits an induced OVA-specific CD8+ T-cell response, slowed tumor growth, and prolonged long-term survival in an OVA-expressing E.G7 thymoma murine model [93]. In addition, GrpE, a cofactor of heat-shock protein 70 (HSP70), promotes Th1-cell-type immunity by interacting with TLR4 located on DCs. These effects of GrpE on DC activation are mediated by the downstream activation of the MyD88, TRIF, MAPK, and NF-κB signaling pathways [94].

Regarding hemocyanins, our data show the involvement of TLR4 in a hemocyanin-mediated proinflammatory response in APCs [98,99]. Mollusk hemocyanins are large oligomeric glycoproteins widely used as adjuvants, peptide and hapten carriers, and nonspecific natural immunostimulants in certain tumor therapies [100,112,113,114]. The biochemical and biophysical attributes of several hemocyanins with immunomodulatory properties purified from wild gastropods, including Keyhole limpet hemocyanin (KLH) from *Megathura crenulata* which has two immunization forms (high-molecular-weight and subunit clinical-grade formulations), CCH from *C. concholepas*, FLH from *Fissurella latimarginata*, RtH from *Rapana thomasiana*, HtH from *Haliotis tuberculata*, and HlH from *Helix lucorum*, have been characterized [100,112,113,114]. The molecular mass of the hemocyanin oligomeric structure in these species is as high as 8 MDa, and each hemocyanin forms a cylinder of approximately 35 nm in diameter known as a didecamer. Each didecamer comprises 20 subunits (approximately 400 kDa each). However, fewer decamers or multimers have been identified. The hemocyanin subunits contain eight globular domains known as functional units (FUs), and each FU includes an active site in which oxygen reversibly binds [115,116,117,118]. Interestingly, hemocyanins consist of one or two types of subunits; for example, FLH is composed of only one subunit, whereas KLH, CCH, RtH, HtH and HlH are each composed of two subunits. Furthermore, these proteins fold into a homodidecamer, such as KLH, or heterodidecamer, such as CCH [101,102,103]. Another essential feature of hemocyanins is an abundant glycan content, which reaches 3–4% *w/w*. In the case of KLH, cryogenic electron microscopy analyses have shown that carbohydrates are localized on the rim of the cylinder wall and on the wall surface of the molecule [116,119]. The N- and O-linked carbohydrates are added glycosylation, and their abundance and heterogenicity vary among species, but are primarily mannose-rich N-glycans and N-mixed carbohydrates with fucose, galactose, GlcNAc, and glycosylation branches that are not found in mammals [98]. Glycosylation is essential for the stability of the hemocyanin oligomeric structure [98,120,121,122]. Furthermore, glycosylation contributes to hemocyanin-induced immunostimulant properties, highlighting the relevance of posttranslational modifications in adjuvant proteins [98,104,121,123].

Notably, we confirmed glycan-dependent binding of hemocyanins to chimeric TLR4 in vitro and in vivo [98,99]. Indeed, DCs from mice with deficient MyD88 expression are partially activated by FLH, suggesting a role played by the TLR pathway in hemocyanin recognition leading to APC activation. Moreover, hemocyanin-induced proinflammatory cytokine secretion is impaired in several models of APCs lacking functional TLR4. Furthermore, we have shown that KLH and FLH induced TLR4-dependent ERK1/2 phosphorylation, a key event in the TLR4 signaling pathway [99].

Plant lectins selectively bind to carbohydrate motifs, and they can be purified by affinity chromatography using different ligands [124,125]. Mistletoe lectin-I (ML-I) and jacalin have been studied as potential mucosal adjuvants because their derivatives bind to mono- or oligosaccharides on mucins or IgA molecules and target the mucosal epithelium [107]. Soybean agglutinin (SBA) is a nonfibrous carbohydrate-related protein with a molecular mass of 120 kDa and the main non-nutritional factor in soybean. Specifically, SBA is a tetrameric N-acetyl-D-galactosamine (GalNAc) and galactose-specific lectin that forms a unique cross-linked composite with a variety of naturally occurring and synthetic multi-antennary carbohydrates with terminal GalNAc or galactose residues [106]. Regarding their mechanism of action, interactions between glycosylated TLR receptors and certain lectin types on APCs has been identified [126]. For instance, SBA stimulates TLR4 in a reporter cell line. This effect was specific for TLR4, and no agonist effect was observed for TLR-2/6, -3, -5, -7, -8, and -9 [106].

A single intratumoral injection of recombinant Mistletoe lectin (Aviscumine) prolongs the median survival of glioma-bearing mice [108]. The Lavelle group showed, for the first time, that ML-1 exhibits high mucosal adjuvant activity when administered with herpes simplex virus glycoprotein D2, showing an enhanced type Th2-cell response to the bystander antigen [109]. This outcome suggests that ML-I provides a platform for the generation of effective mucosal adjuvants due to its ability to penetrate the gut, where it induces mucosal and systemic immune responses [109,110,111].

In the case of Jacalin, early studies have shown its adjuvant effects on humoral immune responses in mice immunized with a lysate or viable epimastigote forms of *Trypanosoma cruzi*, which result in a marked increase in the levels of anti-*T. cruzi* antibodies [124]. However, another plant lectin, ArtinM, administered to mice immunized against neospora, a dog parasite, shows a more significant immunostimulatory and adjuvant effect than Jacalin [127].

### 2.4. TLR5-Dependent Activation by PBAs

The receptor TLR5 responds to flagellin from β- and γ-proteobacteria [128]. The product encoded by the flagellin gene in *Salmonella* binds to TLR5, inducing MyD88-dependent signaling with subsequent activation of the NF-kB pathway in epithelial cells, monocytes, and DCs [129]. Flagellin is the structural component of the flagellum, whose primary function is bacterial motility [130]. The amino acid sequences of the N- and C-terminal regions are conserved among different bacteria. These regions are critical for flagellin oligomerization [131,132]. In contrast, the central domains are highly variable and might contribute to protein stability; the N- or C-terminus of full-length flagellin has been reported to be a safe and nontoxic adjuvant of AB-type toxins in protein form [133,134,135]. However, high doses are related to systemic inflammation and liver injury. Therefore, flagellin has been produced as a recombinant protein and fused or co-expressed with different antigens to boost the immune response mediated by TLR5 [136,137].

The TLR5 ligand flagellin has been studied to determine its usefulness as an adjuvant. The first evidence of flagellin proinflammatory activity was observed with *Salmonella* flagellin, which was shown to be a potent inducer of cytokines at sub-nanomolar concentrations in a promonocytic cell line [138,139,140]. Flagellin can profoundly activate migratory lung DC (migDC) subsets and upregulate CD40, CD80, CD86, and CCR7 in these DCs [141,142]. The adjuvant activity of flagellin has been shown in a mouse models of infectious diseases, leading to adequate protection against infection, indicating that flagellin can be used as a carrier for peptides derived from influenza virus [143] and in an experimental vaccine for *Schistosoma mansoni* [143]. Subsequently, many reports have described flagellin as a vaccine adjuvant and carrier in preclinical studies with several antigens of microbial origin, including vaccinia virus [144], the parasite *Plasmodium falciparum* [145,146] and the HIV gp40 protein [147]. Notably, flagellin has been co-administered with allergens to inhibit airway allergic disease in a murine model of allergic rhinitis [148,149]. Finally, flagellin has been used as an adjuvant in a recombinant hemagglutinin (HA) fusion vaccine (VAX125), inducing higher antibody titers in humans than the vaccine alone [150,151].

*Mycoplasma hyopneumoniae* is the etiological agent of porcine enzootic pneumonia. The P97 protein can mediate microbial adhesion to epithelial cells in the respiratory tract. Remarkably, recombinant expression of the P97 C-terminal domain triggers concentration-dependent TLR5 activation, similar to flagellin, and stimulates the production of IL-8 in HEK-Blue mTLR5 cells. Mice immunized with P97c fused to the ectodomain matrix 2 protein (M2e) of influenza A virus exhibit a high antibody titer against the M2e epitope, which is associated with a hybrid Th1-/Th2-cell immune response [152].

Table 4 summarizes PBAs from bacterial origin with TLR5 adjuvant potential.

### 2.5. Interaction of PBAs with C-Type Lectin Receptors

C-type lectin receptors comprise several families of receptors, including collectins, selectins, endocytic receptors, and proteoglycans, whose interactions with their glycosylate ligands can be calcium-dependent or calcium-independent [156]. The CLRs possess one or more carbohydrate recognition domains (CRDs) or C-type lectin-like domains (CTLDs), which recognize other noncarbohydrate agonists [157]. Upon ligand recognition, antigen is internalized, processed, and presented, inducing intracellular signaling pathways that regulate cellular function [158]. However, few reports on the roles played by CLRs in PBA recognition and modulation of immune responses have been published. Moreover, some studies have suggested that CLRs are dispensable for some PBAs [159,160], whereas for hemocyanins are indispensable [158].

Due to the multivalent nature of their glycosylated residues, hemocyanins, in contrast to other PBAs, interact not only with TLR4 as previously described but also with several CLRs. Indeed, we have previously shown that murine APCs internalize hemocyanins in a glycosylation-dependent manner through receptor-mediated endocytosis with proteins that contain a CTLD, such as the mannose receptor (MR) and macrophage galactose lectin (MGL) [98,99] (Figure 4). Similarly, we have observed that hemocyanins directly bind to CRDs in the MR and DC-SIGN (a DC-specific ICAM-3–grabbing nonintegrin) with high affinity constants, colocalizing with these receptors after being internalized into human DCs through clathrin-mediated endocytosis [104]. Notably, MR lacks a cytoplasmic domain and therefore cannot transduce external signals to intracellular pathways, requiring its cooperation with other innate immune receptors such as TLR4 [99].

Importantly, RtH presents significant adjuvant properties when administered to mice in conjunction with bacterial or viral proteins, a toxoid, and an influenza preparation [161]. In the same antigen preparations, HtH increases anti-toxoid IgG antibodies in the serum of mice to levels comparable to those produced by mice that receive toxoid Al(OH)_3_. In addition, HtH induce a strong anti-influenza cytotoxic response [162]. The FLH shows better immunogenic capabilities than CCH and KLH, exhibiting significant antitumor activity in a B16F10 mouse melanoma model [101] and murine model of oral cancer [163]. The interaction of hemocyanins with various receptors endows them with advantages as adjuvants because they can activate different signal transduction pathways, leading to potent immunostimulatory effects.

## 3. PBAs and the Adaptive Immune Response

Many adjuvants primarily target DCs to induce cellular activation, including antigen presentation. Pattern recognition receptors s enhance the expression of CCR7 on APCs, which promotes their migration to draining lymph nodes, at which time, protein antigens are processed and loaded onto the MHC, facilitating the signaling required to activate naïve antigen-specific Th cells [164]. Upon activation, DCs upregulate the expression of MHC-I/MHC-II and costimulatory molecules and release cytokines and chemokines that polarize T-cells toward acquisition of a Th1, Th2, or Th17 phenotype (Figure 5).

Specifically, IL-12 promotes the acquisition of the Th1 phenotype, primarily contributing to cellular immunity [165,166]. In contrast, IL-4 and IL-10 in the absence of IL-12 promote acquisition of the Th2 phenotype, which stimulates humoral immunity [165,166]. The Th17 phenotype acquisition is promoted by TGF-β and IL-6, which are important to mucosal immunity and protect against bacterial and fungal infection [167]. A specific cytokine profile is required to overcome immune tolerance and is controlled by Tregs [168]. Activated Th-cells upregulate CXCR5 expression, which mediates Th-cell migration to the interface between the B-cell follicle and T-cell areas. Th-cells express IL-21 and CD40 L, which stimulate the clonal expansion of antigen-activated B cells [169]. Antigen-specific B-cells can migrate to the medullary cord and differentiate into short-lived plasma cells. In contrast, other activated B-cells migrate to B-cell follicles and form germ centers (GCs) [170]. In GCs, B-cells differentiate into recirculating memory B-cells or long-lived plasma cells (LLPCs) that migrate to the bone marrow [171]. This process results in affinity maturation during the antibody response, which generates sustained antibody responses [6].

The PBAs can regulate lymphocyte function and adaptive immune responses at different levels. The effect of a PBA on the adaptive immune response is directly related to its immunogenic potential, which is associated with the type of cognate PRR, type of cell, and several tissue-specific factors. For instance, the immunological potency of TLR/CLR agonists has been reported to vary depending on the cell type [73]; that is, certain MPLA formulations behave as full agonists or partial agonists depending on whether they target human or murine TLR4. In addition, most TLR/CLR agonists induce the expression and secretion of proinflammatory cytokines from Th1-/Th2-/Th17-cell types to different degrees and in different timeframes, suggesting differences in the signal transduction pathways that they trigger.

Importantly, antibodies are the crucial effector molecules necessarily induced by vaccines because antibodies trigger immune responses (neutralization, opsonization, complement activation, and antibody-dependent cellular cytotoxicity). Therefore, as thymus-dependent antigens, PBAs, contribute directly or indirectly these immune responses. Furthermore, with the development of bioinformatics tools and information on the sequences of protein antigens, T-cell receptor (TCR) and B-cell receptor (BCR) epitopes can be designed for epitope-based vaccines [172]. This approach endows PBAs with a significant advantage over other types of adjuvants since many structures can be epitopes, and PBAs are the only adjuvants that can directly induce cellular and humoral immunity. Because only MHC-peptide complexes can bind TCRs and activate T-cells, T-cell epitopes in PBAs can be identified or designed to be specific to MHC molecules expressed by a cell. For example, T and B epitopes in flagellin from *B. pseudomallei* have been predicted, and the predicted peptides have been synthesized and characterized using bioinformatics tools. As a result, two of the produced peptides include dominant immunoreactive epitopes, which elicit cytokine production in human peripheral blood mononuclear cells [173]. Undoubtedly, the limitations associated with this type of approach need be considered to prevent antigen-associated complications that may result in adverse effects [174].

## 4. Phase Clinical Studies for Evaluating PBAs as Immunomodulators

Phase clinical studies on potential adjuvant substances are crucial because they form the basis on which future vaccine trials are developed. Our current knowledge regarding the mechanisms of action of PBAs is minimal; therefore, more studies are needed to better understand how PBAs initiate and potentiate immune responses. Nevertheless, several clinical trials have been initiated to evaluate the safety, tolerability, immunogenicity, efficacy, and effectiveness of PBAs in different vaccines against pathogens and cancer [175]. An increase in the number of preclinical and clinical studies based on PBAs may cement their future use in human vaccines.

### 4.1. Flagellin

Clinical trials conducted to test flagellin efficacy as a carrier/adjuvant have been promising. Given the clear immunomodulatory properties of flagellin, recombinant flagellin called Entolimod (CBLB502) has been developed; this preparation is highly purified to a current good manufacturing practice (CGMP) grade and has been shown to be exceptionally stable with lower toxic, and higher immunogenicity than native flagellin [155,176]. Furthermore, Entolimod has been entered into clinical trials as a potentiating adjuvant to promote antitumor responses and reduce radiation- and/or chemotherapy-induced side effects (clinicaltrials.gov accessed on accessed on 14 June 2022 [175] identifiers: NCT02715882, NCT03063736, NCT01527136, and NCT01728480). In addition, CBLB502 protects the liver by increasing the resistance of hepatocytes to acute liver injury via the NF-κB and IL-22 STAT3 signaling pathways [154]. Moreover, CBLB502 has been shown to promote NK-cell-mediated immunity in cytomegalovirus-infected mice [177].

In phase I clinical vaccine trials of a quadrivalent influenza vaccine, which is composed of four HA subunits fused to flagellin, low reactogenicity and a favorable balance of tolerability and immunogenicity were observed in humans [178]. Another phase I clinical trial with healthy adult volunteers has been conducted to evaluate the flagellin/F1/Va vaccine against pneumonic pestis, and the results showed that the vaccine is well tolerated and induces specific antibody activity, which increases as the dose is increased [179]. Moreover, in a phase I clinical trial in patients with non-small-cell lung cancer conducted to evaluate the safety of a DC vaccine pulsed with survivin mucin 1, cell surface associated (MUC1), silenced with suppressor of cytokine signaling 1 (SOCS1), and flagellin used as an adjuvant, the results showed that the vaccine is well tolerated and improves patients’ quality of life [180].

Plague is an infectious disease of animals and humans caused by *Yersinia pestis*, a Gram-negative coccobacillus, that is acute and often fatal. A recombinant protein in which two protective antigens (F1 and V protein) of *Y. pestis* are fused to the hypervariable region of flagellin was designed to protect against respiratory exposure to *Y. pestis* in nonhuman primates [181]. In phase I clinical trials (Clinicaltrials.gov [175] identifier: NCT01381744), increasing doses of the flagellin/F1/V vaccine were administered to healthy volunteers from 18 to 45 years old via the intramuscular route on days 0 and 28. The results from this clinical trial have not been formally posted to date.

Flu is a contagious respiratory illness caused by influenza viruses that infect the nose, throat, and sometimes the lungs. In M2e, a series of 24 nonglycosylated amino acids, has been remarkably conserved in human epidemic influenza virus strains throughout the past century [182]. Therefore, the influenza virus M2e is considered a promising candidate for a universal vaccine. In this context, a vaccine has been developed based on a recombinant protein with four tandem sequences of M2e fused to the C-terminus of flagellin. This vaccine is called VAX102, and it has been entered into in phase I/II clinical trials with the identifiers NCT00921947 and NCT00603811) [175]. The vaccine VAX102 induces a fourfold increase in anti-M2e antibody titers in humans. The effectiveness of VAX102 has also been demonstrated in another phase I clinical trial, in which the vaccine was given to healthy adults previously immunized with trivalent inactivated influenza vaccine (TIV) [183]. The vaccine shows greater immunogenicity and provides cross-protection. In addition, VAX102 is safe and well tolerated [183,184].

Another methodology is based on the outer membrane glycoprotein HA of influenza, the molecular target of the influenza vaccine. Therefore, a panel of influenza vaccine formats in which the HA globular head was fused to *Salmonella typhimurium* flagellin (VAX125) have been evaluated in phase I/II clinical trials (Clinicaltrials.gov accessed on accessed on 14 June 2022 [175] identifiers: NCT00730457). VAX125 induced a fourfold serum hemagglutination-inhibition (HAI) response in more than one-half of the subjects [150]. In another study, VAX125 induced a 10-fold increase in HAI antibody levels in subjects older than 65 years [185]. Thus, a novel H1N1 pandemic influenza vaccine formulated based on three inactivated novel recombinant H1N1 influenza vaccines (VAX128) was constructed [151]. The globular head of the novel H1N1 HA1 domain is genetically fused to the C-terminus of flagellin (VAX128A) and used to replace the D3 flagellin domain (VAX128B) or fused to both positions (VAX128C) in the VAX128 vaccine. In clinical studies, high HAI titers and high seroconversion and seroprotection rates have been observed at doses ≥2.5 μg in young adults (18–49 years old). In contrast, higher doses are required to achieve similar effects in older patients (≥65 years old). Notably, substitution at the D3 position of flagellin position in VAX128B and VAX128C results in a protein that is at least as immunogenic as the modified C-terminal fusion protein in VAX128A and is better tolerated at higher doses. Due to steric effects, receptor binding and/or the molecular weight ratio between HA and flagellin, substituting the D3 domain with the C-terminal fusion may make VAX128C safer and more immunogenic than the other two constructs [151].

Then, VAX2012Q, a quadrivalent influenza vaccine comprising four hemagglutinin subunits fused to flagellin based on VAX128C (H1N1), VAX181 (H3N2), VAX173 (B-YAM), and VAX172 (B-VIC) was studied in a phase I clinical trial [178]. All four components of VAX2012Q elicited seroprotective immune responses against the respective influenza type A or B virus. In addition, no safety concerns have been reported for VAX2012Q.

### 4.2. Toxoid Adjuvants

In this section we highlight ganglioside ligands of TLR2 which are essential for immune responses to cholera and influenza vaccination. The oral cholera vaccine Dukoral (with inactivated whole *V. cholerae* O1 bacteria plus recombinant cholera toxin B subunit [rCTB]) shows immunogenicity, is safe, and is well tolerated in people [186]. In phase I/II clinical trials, a vaccine (based on specific peptides of the HSP60 protein together with rCTB) against Bechet’s syndrome is also well tolerated and safe [187].

A LTh(AK)-adjuvant vaccine is a detoxified derivative of *E. coli* heat-labile toxin and inhibit ADP-ribosylating enzyme activity. An intranasally administered inactivated influenza vaccine with LTh(AK) did not cause effects on the central nervous system, suggesting a high safety profile in humans [188]. In addition, the adjuvant LTh(AK) vaccine leads to significantly enhanced mucosal immunity in specific IgA. Two intranasal vaccinations tested in a phase I (NCT03293732) and phase II (NCT03784885) clinical trial [175,188] are tolerated. Hence, bacterial mutant derivatives of *E. coli* heat-labile toxin and cholera enterotoxins with appropriate nontoxic phenotypes have been developed [189,190]. A phase II clinical trial with an oral inactivated enterotoxigenic *E. coli* vaccine for children (ETVAX), which incorporates the double-mutant heat-labile toxin (dmLT), shows safety and high immunogenicity, with antibody levels increasing significantly between the baseline to postimmunization measures [191].

### 4.3. Hemocyanin and Antitumor Vaccines

Antitumoral vaccines are intended to treat existing cancer by strengthening the body’s natural defenses against cancer and have emerged as alternatives to antiproliferative treatments, such as chemo- or radiotherapy. Hemocyanins have been used as carriers of tumor-associated antigens, protein adjuvants for DC-based therapeutic cancer vaccines, and nonspecific immunostimulants in immunotherapy for superficial bladder cancer. In the 1960s, the outstanding immunostimulatory properties of KLH in animal models and humans was initially reported [192,193], and it has since been assessed for its immunocompetence in human [194]. Keyhole limpet hemocyanin has been used as a hapten carrier to produce antibodies against small molecules, and rapidly extended its immunological and biomedicinally applications [115]. In addition, KLH has been used as a model antigen to study thymus-dependent responses mediated by CD4+ T-cells. It has also been used as an adjuvant in immunotherapy mediated by DCs in which self-tolerance to tumor antigens is disrupted [195]. Therefore, KLH has been considered an ideal immunization antigen candidate for immunotoxicology research. The following properties of KLH have made it a valuable adjuvant: (1) It is available as a clinical-grade product; (2) is highly immunogenic in humans with different ethnic backgrounds; (3) induces potent primary immune responses following a single immunization without adjuvants; and (4) produces quantifiable immune responses that can assessed with validated immune assays to detect changes in immunomodulation. However, immunization and immunoassay protocols still need to be standardized [196].

The benefits of using KLH for treating transitional cell carcinoma, which includes low toxic side effects, were discovered serendipitously [197], opening the opportunity for the use of KLH as a nonspecific immunostimulant to treat this type of cancer [198]. Hence, a vast amount of information on different biomedical applications of KLH includes data on its use as a carrier in vaccines against pathogens and certain cancers and as an adjuvant in DC-based cancer immunotherapy [101,195]. Notably, information on 151 clinical trials registered with the NIH (completed, terminated or recruiting) is currently available [199]. Sialyl-Tn (STn) is the tumor antigen most frequently expressed in breast, ovarian, colon, rectal, stomach, and pancreatic adenocarcinomas. Theratope is a formulation in which STn is conjugated to KLH, and it can induce significant of IgM and IgG antibodies against STn, as measured in titers [200]. A double-blind randomized phase III multicenter clinical trial enrolled 1028 women with metastatic breast cancer with either no current disease or stable disease evidence. The results showed that the Theratope vaccine induces antibody immunity development directed against STn, but no overall beneficial effect on tumor progression or survival has been reported [201].

High levels of the gangliosides GD3, GD2, and GM2 are evident on the cell surface of malignant melanomas, with GD3 being the most abundant. Patients with American Joint Committee on Cancer (AJCC) stage III or IV metastatic malignant melanoma and who were free of detectable melanoma within a period from 2 weeks to 6 months after surgery were vaccinated with a GD3-lactone-KLH conjugate plus immunological adjuvant QS-21. The GD3-lactone-KLH vaccine resulted in the production of IgM and IgG antibodies against GD3 and tumor cells expressing GD3 in most immunized patients [202]. Unfortunately, in a phase III clinical trial, the GM2-KLH/QS-21 vaccination did not improve the outcome for patients with stage II melanoma [203]. For patients with biochemically measured relapsed prostate cancer, a vaccine containing 6 KLH conjugates targeting GM2, Globo H, Lewisy, Tn, TF, and MUC1, respectively, has been developed. All 30 patients in a clinical trial showed significant elevation in titers of antibody against at least two of the six antigens and was proven to be safe [105].

Long-chain polysialic acid (polySA) is a side chain on embryonal neural cell adhesion molecules that is found only in adults with small-cell lung cancer (SCLC). Six patients were vaccinated with polySA conjugated to KLH, and QS-21 treatment resulted in a consistent high titer antibody response [204]; however, peripheral neuropathy and ataxia were detected in several vaccinated patients. Then, in a clinical trial, the lowest optimal dose was determined, and the safety of the NT-polySA vaccine in inducing antibodies was confirmed. Vaccination with NP-polySA–KLH in consistent high-titer antibody responses [205].

The increased use of KLH has led to interest in finding previously undiscovered hemocyanins with similar or better biochemical and immunological properties. Several novel hemocyanins obtained from different organisms have been described in the past few decades, including hemocyanins from *C. concholepas* (CCH) [102], *R. thomasiana* (RtH) [206], *H. tuberculata* (HtH) [207,208], and *F. latimarginata* (FLH) [101]. Notably CCH has been used as a carrier of peptides and hapten molecules and its immunomodulatory properties as a nonspecific immunostimulant in murine models of superficial bladder cancer, melanoma, or oral cancer and indicating that CCH exerts antitumor effects like those of KLH.

A tumor antigen-presenting cell (TAPCell) vaccine based on a melanoma cell lysate as the antigen source called TRIMELVax induces immune responses and increases melanoma patient survival. The TRIMEL has been co-injected with CCH as an adjuvant in castration-resistant prostate cancer patients. The CCH induces an immune memory response in the patients, as measured by delayed-type hypersensitivity (DTH) assay, and no toxic or allergic reactions associated with its subcutaneous administration has been observed [209]. The treatment of CRPC patients with a lysate-loaded TAP Cell vaccine with CCH as an adjuvant is safe and can activate prostate cancer cell lysate (PCCL)-specific T-cells [209]. Mature monocyte-derived circulating idiotype protein (Id)-pulsed DCs and KLH have been administered to patients with stage-I multiple myeloma. The DC-based Id vaccination induces specific T-cell responses in stage-I myeloma patients, as indicated by proliferation assays and T-cell-mediated cytokine release measurements taken after Id stimulation [210].

The hemocyanin RtH exhibits important adjuvant properties when administered to mice in conjunction with either bacterial or viral proteins, a toxoid, and an influenza preparation [161]. When the same antigen preparations are administered, HtH can increase anti-toxoid IgG antibodies in the sera of mice, and the increase is comparable to mice that receive toxoid Al(OH)_3_. In addition, HtH showed a strong anti-influenza cytotoxic response [162]. The FLH has shown better immunogenic capabilities than CCH and KLH, exhibiting significant antitumor activity in the B16F10 mouse melanoma model [101] and in a murine model of oral cancer [163].

## 5. Challenges to the Development of Recombinant PBAs

Recombinant proteins do not present several of the potential risks related to the use of vaccines and adjuvants derived directly from natural sources, therefore, they are promising candidates. Commonly, the four systems most used to express recombinant proteins are: *E. coli*, *P. pastoris*, baculovirus/insect cell, and mammalian cells.

*Escherichia coli* has been extensively used to produce recombinant proteins because its genetic profile is easily manipulated and the cells grow quickly, reaching high density in a short time, and the requirements for bacteria culture are relatively inexpensive [211]. However, the expression of recombinant PBAs in *E. coli* is not easy because some PBAs have a large size which difficult heterologous expression and the presence of posttranslational modifications (i.e., glycosylations). Bacteria lack of the posttranslational modifications machinery and their folding system is not able to process large proteins. Indeed, in the case of glycosylated proteins, posttranslational modification is a critical attribute of the biopharmaceutical product, as glycosylation affects the efficacy, serum half-life, and antigenicity of a molecule. Moreover, prokaryotic chaperons may induce folded protein intermediates that differ from those produced in a eukaryotic folding pathway. In addition, the intracellular *milieu* of a host prokaryotic cell is different (pH, redox potential, and ion content differences), affecting the folding of recombinant proteins [212].

High expression of recombinant proteins can lead to aggregates within inclusion bodies. Therefore, they can be purified under denaturant conditions with chaotropic agents and reducing agents. Following denaturation, several techniques are used for protein refolding and recovery of protein biological activity, such as dilution, dialysis, and chromatography. Another recurrent issue is the presence of LPS in the batch of purified recombinant proteins. Lipopolysaccharide is a TLR4 agonist that can act as an adjuvant. However, several commercially kits can be used to remove LPS contamination. This allows that some recombinant PBAs can be produced under GMP conditions and consequently be used in a broad range of basic and clinical studies.

### 5.1. Recombinant Bacterial PBAs

Several components are required to express heterologous genes in prokaryotic cells, such as *E. coli* cells. The design of the vector, including the appropriate promoter, is essential to ensure efficient protein expression. Another important component is the transcription terminators added to the expression vectors, that helps to stabilize the mRNA, thus increasing the protein expression levels. In addition, the expression vectors need to contain a translational initiation site that corresponds to the Shine-Dalgarno sequence in the prokaryotic system [213].

The expression system based on the bacteriophage T7 RNA polymerase can direct high levels of transcription regulated by the T7 promoter, which can lead to high levels of recombinant protein accumulation. In fact, this system saturates the translational machinery of *E. coli*; therefore, the rate of protein synthesis from an mRNA depends primarily on the efficiency of its translation, which can lead to an accumulation of the total cell protein that is higher than 50% [214,215]. Indeed, an issue associated with a high protein expression level is the formation of inclusion bodies with proteins that are inactive because of misfolding. The inside of a protein has hydrophobic regions, but due to overexpression, these hydrophobic regions can interact with each other, leading to protein aggregation [216].

Among PBA candidates, one of the most commonly used adjuvants in vaccines is flagellin. Large-scale production of recombinant full-length and fusion flagellin constructs has been successfully achieved with *E. coli*. However, because flagellin binds TLR5 and because this receptor is widely expressed in the epithelium, flagellin can potentially induce systemic adverse reactions due to its overt activation of TLR5 [217]. Therefore, novel recombinant flagellin with deletions in domains D2 and part of D1 (critical for preventing adverse reactions but can trigger TLR5) that exerts significant effects on antigenicity and immunostimulatory adjuvant activity have been developed [218].

Although porins produced by the *S. enterica* serovar *Typhi* play roles in diagnostics and vaccination, conventional purification methods can be tedious and time-consuming; therefore, an easy and rapid purification method that can be upscaled is needed. To address this need, OmpF and OmpC from *S. typhi* have been cloned into the pQE30UA vector and expressed in *E. coli.* Recombinant porins are then solubilized in 8 M urea and purified by immobilized metal affinity chromatography (IMAC). The protein yields with this system are 30 and 25 mg/L recombinant OmpF and OmpC, respectively. Importantly, an increase in anti-OmpF and anti-OmpC IgG antibodies has been observed in mice immunized with the recombinant proteins [219].

Ply from *S. pneumoniae* has been shown to interact with TLR4 and shows adjuvant efficacy. However, it is highly toxic [52], which has been a major problem for the incorporation of Ply into new vaccines. By producing recombinant proteins, regions or domains that can induce toxicity can be removed, and fusion proteins that enhance immunity can be generated. Studies have demonstrated that recombinant Ply that has been mutated through site-directed mutagenesis shows lower toxicity than wild-type Ply. A single mutation (DA146 Ply) prevents the protein from lysing erythrocytes and nucleated cells or forming pores [220]. Therefore, this mutant has been suggested to be a subcutaneous immune adjuvant.

A recurrent issue is the presence of LPS in the batch of purified recombinant proteins. Lipopolysaccharide is a TLR4 agonist that can act as an adjuvant. Several commercially available kits can be used to remove LPS contamination to a final concentration less than 0.1 EU/µg. In the case of DnaJ-ΔA146Ply, LPS removal significantly enhanced DnaJ-specific serum IgG and saliva sIgA antibody responses, providing protection against pneumococcal infections [88,221].

The SIP gene from GBS has been cloned and introduced into *E. coli* strain BL21-CodonPlus (DE3). Our group reported that optimization of the SIP purification process reduces the amount of impurities and improves the immune response against vaginal GBS infection [84].

### 5.2. Recombinant Hemocyanin

No whole recombinant hemocyanins are currently available, possibly due to their high molecular mass and complex quaternary structure. Recombinant expression of 19 different substructures in KLH has been obtained [216]. In this case, the authors used an *E. coli* system to express the functional unit 1 h (FU-1 h) of KLH1, as confirmed by Western blotting. However, other FUs cannot be expressed. The comparison of the two-dimensional immunoelectrophoretic patterns and absorption spectra of two KLH1-h preparations (native versus recombinant FU-1 h) may explain the lack of biological activity of recombinant KLH1-h. Although each FU of the KLH2 had been successfully expressed in *E. coli*, immunogenicity has not reported [216].

## 6. Pharmaceutical and Relevant Regulatory Considerations in the Assessment of PBA Safety

### 6.1. Vaccine Administration System

Studies have been conducted to improve the quality, safety, and efficacy of vaccines administration to patients. Historically, the parenteral routes of intramuscular and subcutaneous injection have been most commonly used routes of vaccine administration. However, these routes present certain disadvantages, such as pain at the puncture site, increased infections due to reused contaminated needles, psychological rejection by patients, and poor immunogenicity compared to that of vaccines administered through the mucosal route [222]. Specifically, vaccines administered through the mucosal route show higher immunogenicity than those administered through the parenteral route, and therefore, vaccines formulated for mucosal administration vaccines have garnered considerable interest. However, few mucosal vaccines have reached the market; therefore, the development of mucosal vaccines remain a pharmaceutical goal [223]. It should be emphasized that of the several factors which determine the efficacy of a vaccine, the route of administration is an important vaccination parameter that can substantially modify the generation and profile of the immune response [224]. Thus, the choice of the route depends on the infections disease in particular on the route of infection and transmission [222]. For several diseases, mucosal vaccines cannot be developed, and intradermal, subcutaneous, or intramuscular vaccines are an option.

### 6.2. Safety of Adjuvants: Regulatory Framework

Given the lack of knowledge and legislation, international regulation of adjuvants remains a challenge. In 2003, the WHO published a guide for nonclinical recommendations for the production of vaccines combined with new adjuvants, pointing out certain pharmacological, toxicological, and other criteria that must be met. However, this document does not distinguish between different types of adjuvants for their evaluation, nor does it indicate the specific trials to be carried out to demonstrate their safety. Indeed, PBAs must meet the exact requirements of other adjuvants, although have entirely different characteristics. Moreover, no assays or other explicit requirements are recommended to evaluate the safety of an adjuvant in a vaccine, but it is prudent to characterize minimum and maximum doses and to perform a battery of genotoxicity studies in preclinical studies and during Clinical Trial I [225,226]. Notably, when evaluating a vaccine for safety and efficacy, the FDA consider an adjuvant as a component of the vaccine; therefore, it is not licensed separately.

## 7. Advantages/Disadvantages of PBAs Compared to Conventional Adjuvants

Safe and effective vaccine adjuvants that can be used to promote protective immunity to prevent infectious diseases and/or reduce deleterious immune responses, as in cases of autoimmune diseases, are lacking. Therapeutic vaccine approaches can be classified into those targeting conditions in which antibody responses can mediate protection and those in which the principal focus is promotion of effector and memory cellular immunity. For example, traditional aluminum adjuvants can trigger strong humoral immune reactions but weak cellular immunity, limiting their application to certain vaccines. Importantly, PBAs can elicit strong cell-mediated immune responses to administered antigens, thereby showing an essential advantage over currently adopted adjuvant approaches [222,227,228]. The PBAs present several advantages over nonprotein-based adjuvants in development, which are summarized with examples in the following subsections.

### 7.1. Natural Biocompatible and Biodegradable Polymers

Biocompatibility, defined as the ability of a substance to induce an appropriate host response in through specific application [229], is the priority when selecting adjuvants to ensure minimal toxicity and injury potential while maximizing physiological/immunological reactivity. In this context, PBAs efficiently enhance immunological mechanisms due to their biocompatibility and biodegradability. The PBAs show lower risks of inducing acute local or systemic immunotoxic reactions and delayed adverse side effect than other adjuvants [11,230]. Notably, hemocyanin and specifically subcutaneously administered KLH in its two immunization forms (high-molecular-weight and subunit clinical-grade formulations) have been well tolerated in human clinical settings, with only mild effects reported; therefore, they have an excellent safety profile [196]. Similar results have been obtained with CCH [209].

Another essential feature of PBAs is biodegradability. Indeed, PBAs are proteins and, therefore, are polymers that can be broken down into biologically compatible molecules, usually via catalysis by enzymes involving both hydrolysis and oxidation [231]. In addition, PBAs exhibit better shelf-life stability than other adjuvants, which makes them more versatile in delivery strategies. For instance, specific PBAs, such as bacterial-derived toxins, are gold-standard mucosal adjuvants and can efficiently induce immune responses, including secretory IgG and IgA production [232,233].

### 7.2. Minimal Reactogenicity and Toxicity

Evidence strongly suggests that certain mechanisms critical for the immunostimulant effects of adjuvants are also involved in the induction of adverse effects [234]. Indeed, reactogenicity produces excessive immunological responses, either locally (swelling and pain at the injection site) or systemically (fever and general malaise). Examples of reactogenic adjuvants include saponins and oil emulsions, whose inflammatory adjuvant action involves increased vaccine reactogenicity leading to improved immunogenicity. Hence, reactogenicity can be prevented only when it the inflammatory effect can be separated from the adjuvant action [235]. Different methods based on structure–activity relationship studies of immune agonists have been developed, including chemical modifications, genetic engineering, and synthetic biology, allowing immunogenic adjuvants with minimal toxicity to be obtained [11,236]. For example, the A subunit of the *E. coli* H10407 enterotoxin shows powerful adjuvant properties in poultry; however, it induces toxicity in mammals. To neutralize the toxic effect, two amino acids in this protein were genetically changed, not affecting its adjuvant efficacy [237].

Moreover, PBAs can be co-delivered with antigens through gene fusion to obtain an adjuvant–antigen codelivery vaccine, whose reactogenicity depends on its configuration, as in the case of bacterial flagellin. Flagellin is genetically fused with the HA1 fragment of influenza A virus (A/Solomon Islands/3/2006 [H1N1]) in the C-terminus or the D3 domain or at both the C-terminus and N-terminus to obtain VAX128A, VAX128B, and VAX128C. The phase I clinical trials with VAX128A and VAX128B have been stopped because at high doses (up to 16 µg), but not low doses, patients presented with flu-like symptoms. Moreover, among these vaccines, VAX128C shows lower reactogenicity but retained its immunogenicity [151]. In addition, another approach for reducing adjuvant toxicity is synthetic biology, which is based on computational tools and mathematical modeling analysis of multiple datasets to predict the immune responses of animals and humans to adjuvants [238]. This approach has been used to produce monophosphoryl lipid A analogs, which show potent adjuvant activity and induce low toxicity [236]. It is also beneficial for optimizing PBAs. Indeed, because of its large size, flagellin produced in truncated forms as fusion proteins for use in experimental vaccines against urinary tract infections shows higher affinity than the full-length protein for TLR5 in silico and in vivo; thus, the truncated forms used in vaccines induce potent cellular and humoral immune responses [239].

### 7.3. Binding to Innate Immune Receptors

The engagement of innate immunity receptors on the surface of APCs increase the endocytosis rate of ligands bound to them, quantitatively and qualitatively modulating subsequent adaptive immune responses [240]. Therefore, creating a physical link between an antigen and a PBA agonist of an innate immune receptor using bifunctional reagents or genetic engineering can ensure antigen-PBA codelivery not only in the same cell but also in the same endocytic pathway for antigen processing and presentation with MHCs. This enhances antigen presentation and promotes antigen transport to draining lymph nodes. Furthermore, PBAs can induce the expression of signaling molecules and the secretion of proinflammatory cytokines that tailor the immune response. Therefore, APCs stimulated by PBAs can activate naïve T-cells and induce their polarization toward a Th1, Th2 or Th17 phenotype [37,89,93,233,241]. These effects are particularly relevant for the development of vaccines to prevent infectious diseases that rely heavily on cell-mediated immunity. For instance, CD4+ T-cell immunity, particularly Th1- and Th17-cell-mediated immunity, is paramount for combating fungal infections. However, despite the substantial global burden of these infections, no antifungal vaccines are currently licensed for use in humans [242].

Coupling antigens to PBAs can effectively increase antigen-specific immune responses and reduce the number of off-target effects. For example, an adjuvant codelivery system produced by the fusion of the HSP-70 and HIV-1 p24 proteins stimulates stronger humoral and cellular immunity than a simple antigen-adjuvant mixture [243]. The fusion-based PBA has been associated with more effective delivery to early endosomes and increased expression of activation markers in BMDCs [243]. Another study showed that tumor bearing-mouse mice receiving HSP-70 fused to a tumor antigen show significantly delayed tumor growth and longer survival compared to those receiving a simple adjuvant-antigen mixture [244,245]. Cytokines, such as IL-2, particularly those co-delivered with antigens through gene fusion, induce increased humoral and cellular responses to different antigens in mouse models [222,246]. Furthermore, PBAs, including bacterial-derived toxins, have been used in conjunction with other adjuvants, such as oligonucleotides, glycolipids, and polysaccharides, to exhibit more effective and long-lasting immune responses than either adjuvant delivered alone [247,248,249,250].

However, several factors influencing immunogenicity need to be thoroughly considered when using PBAs as carrier proteins. For example, the hapten, linker, hapten load (number of haptens per carrier molecule), degree of conjugate aggregation, presence of adducts, tertiary structure, dosage, and route of administration, among factors, need to be precisely determined [150,185,247,251,252]. Furthermore, site-directed modification of a protein can be employed when immunogenicity is compromised. For instance, Sortase A, in a class of enzymes found in Gram-positive bacteria, is associated with protein modification based on its recognition of specific protein sequences (LPXTG motifs) and has been successfully used in several mouse models, showing enhanced cellular and humoral immunity [253,254,255].

### 7.4. Natural and Synthetic Vaccine-Adjuvant Sources

In terms of raw materials, synthetic adjuvants are generally preferable over adjuvants derived from animal sources because of their purity, sustainability, and safety. Moreover, specific concerns related to use of PBAs, particularly hemocyanins, have hampered their application. These glycoproteins are derived from natural sources; therefore, they present with variations, resulting in differences in their immunomodulatory activity. However, it is important to highlight that conventional adjuvants such as aluminum salts also exhibit significant variability in their potential immunostimulatory effect, depending on their physicochemical properties, which include particle shape and size, crystallinity, and chemistry [256,257,258]. Therefore, more studies are needed to understand the variability in the immunogenicity of PBAs derived from natural sources.

## 8. Opportunities for PBA Application in Vaccines

Prevention of infectious diseases in elderly people has become a priority as the proportion of older individuals increases globally. The PBAs offer opportunities for the development of adjuvants for vaccines targeting elderly individuals. Innate and adaptive immune responses change with age and grow increasingly aberrant over time [259,260]. These changes are associated with reduced efficacy of vaccines [261]. For instance, the effectiveness of vaccines against influenza-like illnesses in elderly people is only 23% [262]. Moreover, given the impairment of TLR function in older populations, a proper adjuvant capable of inducing a robust immune response is needed to produce a more effective vaccine [259]. Notably, in contrast to other TLRs, TLR5 function and expression are preserved in the monocytes of older individuals [263,264], suggesting that adjuvants targeting TLR5 may be particularly beneficial in vaccines developed for elderly individuals, as shown in older mice successfully vaccinated with a fusion protein of flagellin and hemagglutinin domain 1 [265]. However, enhancement of the immunogenicity of vaccines for elderly people and other susceptible individuals (immunosuppressed patients, infants/neonates, and those suffering from chronic diseases) may not always be desirable, as the increased immunogenicity can cause immunopathology with outcomes that are more harmful than the primary disease. For instance, most flu-related deaths are caused by secondary infections, including bacterial and fungal infections [266]. Therefore, developing effective adjuvants that can fine-tune the immune response of vaccines is particularly relevant when targeting vulnerable people.

Few mucosal vaccines for humans are currently licensed (eight oral vaccines are being used against cholera and Salmonella, poliovirus, and rotavirus infection and one intranasal vaccine is used for treating influenza virus infection), mainly due to the lack of safe and effective mucosal adjuvants for clinical applications [62,267,268]. In addition, there significant advances have been made in injectable vaccines due to the tremendous impact of mRNA-based vaccines and plasmid DNA vaccines. These technologies have yet to be incorporated into mucosal vaccine production, which are currently formulated based on live attenuated and inactivated whole-cell preparations [62]. Due to the ability of PBAs to engage innate immune receptors expressed by APCs and mucosal epithelial cells, such as TLRs (e.g., flagellin, SIP), CLRs (e.g., hemocyanins), and NOD2 (e.g., CT), PBAs can generate a robust anti-PBA immune response that can simultaneously enhance the immune response against the vaccine antigen through a bystander effect. In addition, PBAs can undergo chemical or genetic engineering modifications and be used as delivery vehicles/adjuvants. Notably, the manufacturability of most PBAs using well-established expression systems, purification, formulation, and storage conditions for optimal stability contributes to their effectiveness, acceptable levels of tolerability and safety.

Many questions worthy of continued research on PBA remain, including the search for relevant receptors or pathways. Answers require a wide range of studies on the mechanisms of action of PBAs in the immune system, which may lead to their incorporation into human vaccines. Furthermore, for the use of PBAs as antitumor agent, distinct mechanisms of action of PBAs in normal and cancerous cells need to be considered. Resolving these issues is essential to provide a solid basis for using PBAs as effective adjuvants in the future.

## 9. Conclusions

For the new generation of vaccines against emergent and re-emergent pathogens, either viruses, bacteria, fungi, or parasites, formulated with recombinant proteins, as well as RNA and DNA vaccines, a suitable adjuvant is necessary. Therefore, the selection of an effective adjuvant is crucial. However, despite considerable efforts to develop effective adjuvants, progress has been limited because most adjuvants cause multiple side effects. Thus, the discovery of effective adjuvants is the main challenge to vaccine development. The ideal adjuvant must meet several requirements. In addition to helping confer an intense and long-lasting immune response, the adjuvant must not induce local or systemic adverse reactions. In addition, it must be applied via different routes, including mucosal routes, in children and elderly people and must be effective against weak antigens (e.g., conjugates). Satisfying these criteria is vital to reduce the antigen quantity necessary and to allow multiple doses of vaccine when antigens are presented in low abundance.

Many unanswered questions regarding the mechanisms of PBA action remain, and the search for undiscovered receptors or pathways engaged by PBAs is ongoing. Furthermore, considering their use as antitumor agents, PBAs might follow different mechanisms of action in normal and cancer cells. To provide a solid basis for using PBAs as effective adjuvants in human vaccines, these issues need to be addressed.

In this review, we have described PBAs showing the most promise for use in clinical and preclinical studies and that can undoubtedly be adjuvants for novel prophylactic and therapeutic human vaccines in the future. We propose that PBAs can be exploited in the development of vaccines against pathogens that have proven to be challenging, including intracellular pathogens (*M. tuberculosis*), those with complex life cycles (*P. falciparum*), those that induce immune dysfunction in the host (HIV), those that target immunocompromised individuals (fungi), those with a latent disease phase (herpes viruses) and those that undergo continuous evolution (influenza and SARS-CoV-2).

## Figures and Tables

**Figure 1 pharmaceutics-14-01671-f001:**
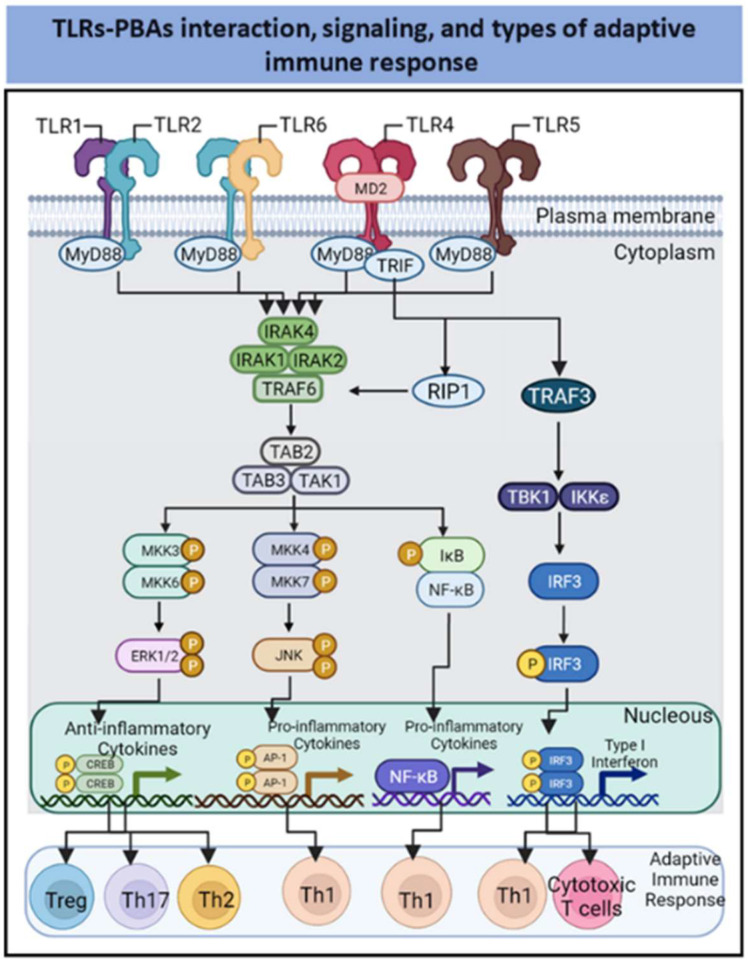
**Molecular targets of TLR-PBAs and downstream signaling leading to different types of immune****responses**. The PBAs interact at the cell surface of APCs with TLR2 (e.g., porins FomA and MOMP, AB-type toxins and OMP 16); TLR4 (e.g., HMGB1, heat-shock-70-like protein 1, and SIP), pneumolysin, hemocyanin ML-I); TLR5 (e.g., flagellin); and heterodimers TLR2–TLR1 (e.g., porin OmpU) or TLR2–TLR6 (e.g., porin MOMP). The TLR signaling is initiated by the dimerization of receptors, leading to the engagement of Toll/interleukin-1 receptor (TIR) domains with MyD88 or TIR domain-containing adapter-inducing IFNβ (TRIF). Engagement of MyD88 recruits downstream signaling molecules to form Myddosomes, which are based on MyD88 and include IRAK4 and IRAK1/2. The IRAK1 activates the E3 ubiquitin ligase TRAF6 to synthesize K63-linked polyubiquitin chains, leading to the recruitment and activation of the TAK1 complex. The TAK1 phosphorylates the canonical IKK complex, ultimately leading to the activation of the NF-κB transcription factor. The TAK1 induces to the activation of MAPKs, including MKK4/7 and MKK3/6, which are further activated through phosphorylation by JNK and p38, respectively. The IKKβ leads to the activation of MKK1 and MKK2 and subsequently ERK1/2. These MAPKs induce the activation of the transcription factors CREB and AP1, which cooperate with NF-κB to promote the induction of proinflammatory cytokine expression. Engagement of TRIF recruits TRAF6 and TRAF3. The TRAF6 can recruit the kinase RIP1 and activate the TAK1 and IKK complex, leading to the activation of the NF-κB and MAPK pathways. The TRIF also promotes TRAF3-dependent activation of TBK1 and IKKϵ, which phosphorylates IRF3. The IRF3 induces type I IFN expression. Depending on their interaction with particular TLRs, PBAs induce the activation of different T-cells profiles. Adapted from Duan et al. [40].

**Figure 2 pharmaceutics-14-01671-f002:**
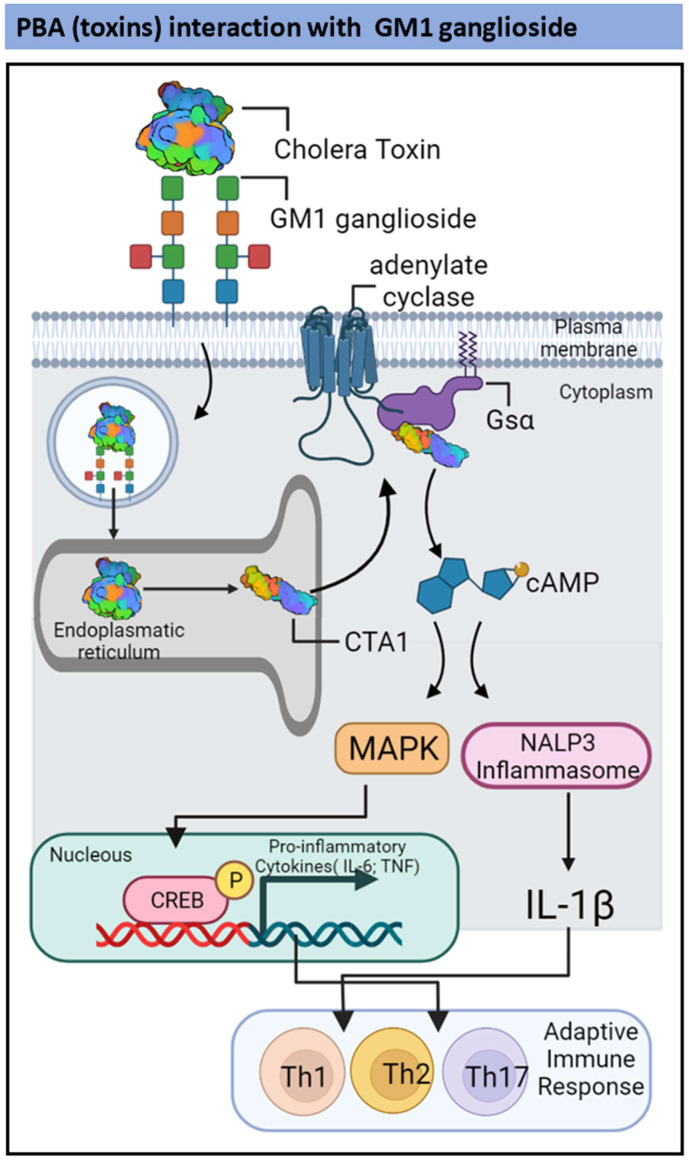
**Molecular targets of protein toxins and the adaptive immune response associated with the ganglioside GM1.** Cholera toxin (CT) can bind with the ganglioside GM1 on the cell surface of APC through CT subunit B (CTB) and be internalized into the endoplasmic reticulum (ER). Within the ER, a disulfide bridge of in the CTB-CTA2 complex is reduced, releasing the CTA1 chain. This chain is then translocated to the cytosol, where it binds to Gsα, which activates adenylate cyclase (AC) to produce intracellular cAMP, MAPKs and the NALP3 inflammasome. As a mucosal adjuvant, CT stimulates innate immune cells to induce Th1-, Th2-, and Th17-cell adaptive immune responses, including antibody (Ab) production and T-cell responses systemically and specifically in mucosal compartments.

**Figure 3 pharmaceutics-14-01671-f003:**
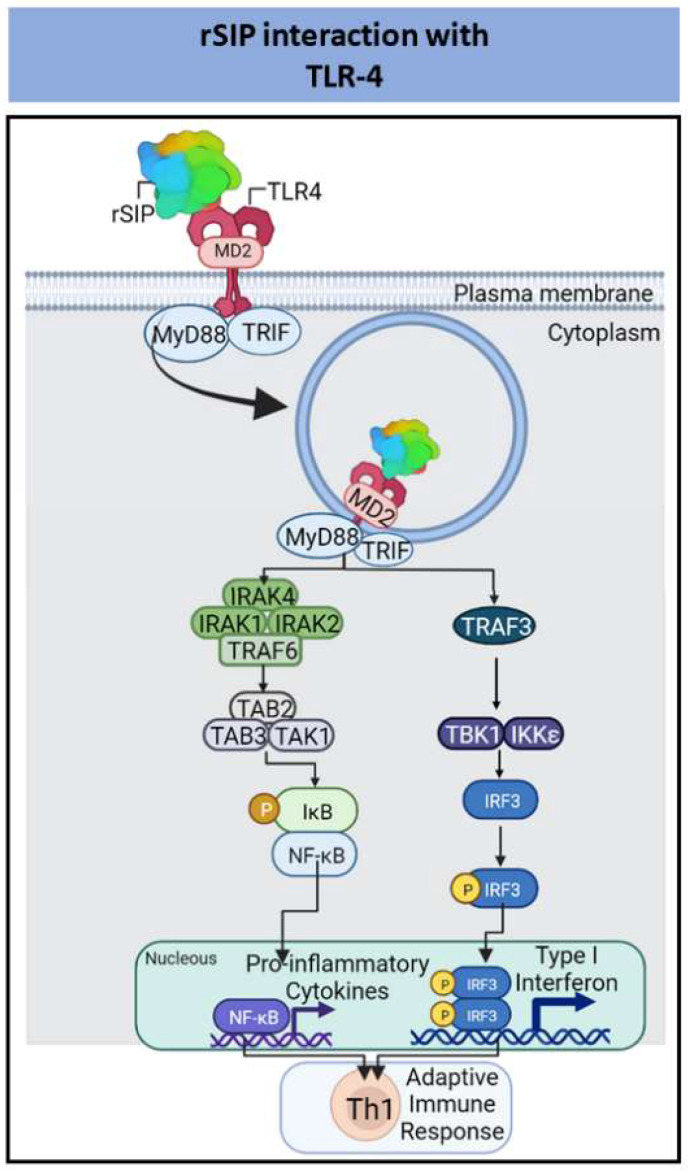
**Molecular targets of rSIP and the adaptive immune response associated with TLR-4.** The model shows that rSIP can interact at the cell surface of APC with TLR4 and induce a TLR4 pro-inflammatory response through the recruitment of adaptors MyD88- and TRIF-downstream signaling pathways, as described in Figure 1. As an adjuvant, rSIP stimulates innate immune cells to induce Th1-cell adaptive immune responses.

**Figure 4 pharmaceutics-14-01671-f004:**
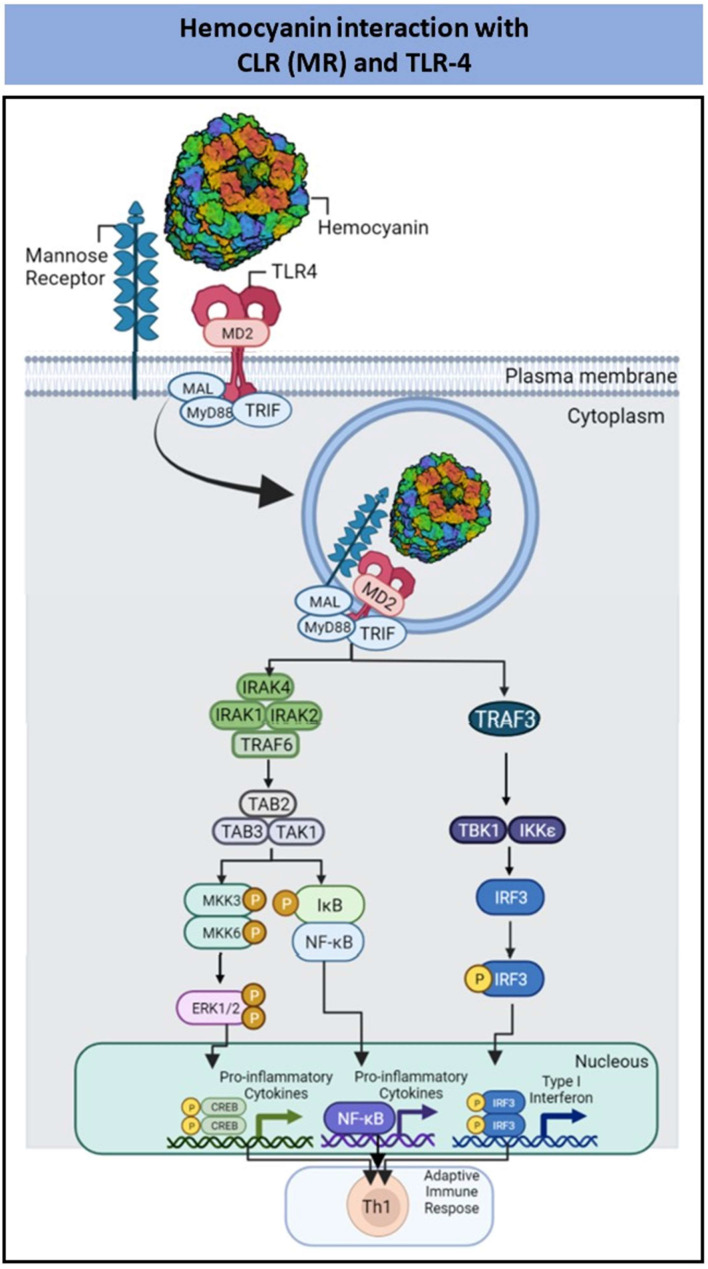
**Molecular targets of hemocyanin and the adaptive immune response associated with C-type lectin receptor and TLR-4.** The model shows that hemocyanins cooperate with mannose receptors and TLR4 at the cell surface of APC, to initiate a proinflammatory response. Hemocyanins activate TLR4 to stimulate MAPK-dependent downstream signaling pathways, as shown in Figure 1. In addition, hemocyanins are endocytosed and activate the Toll/interleukin-1 receptor (TIR) domain-containing adapter-inducing IFNβ (TRIF) pathway. As an adjuvant, hemocyanin stimulates innate immune cells to induce Th1-cell adaptive immune responses.

**Figure 5 pharmaceutics-14-01671-f005:**
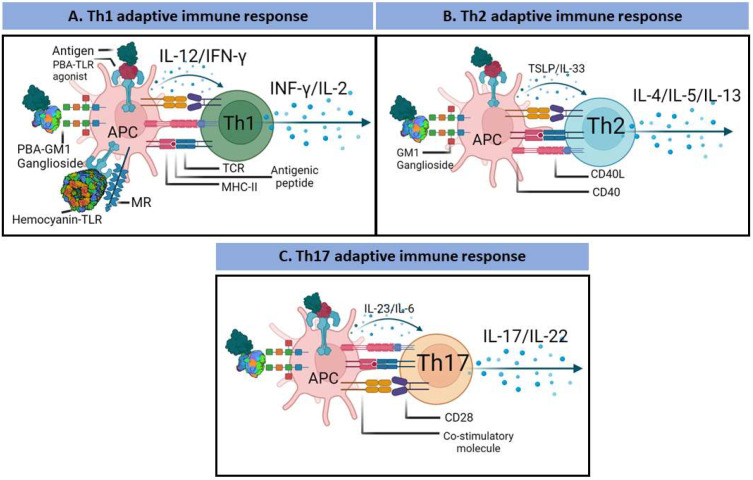
**Mechanism of the adjuvant action of PBAs as vaccine immunomodulators.** The PBAs can interact with APC. Through these interactions, PBAs trigger different mechanisms of the adaptive immune response, which facilitates the polarization of lymphocytes toward Th1-, Th2-, and Th17-cell phenotype acquisition, with subsequent release of their respective cytokines.

**Table 4 pharmaceutics-14-01671-t004:** PBAs with TLR5 adjuvant potential.

Specie	PBA	Innate Immune Response	Adaptive Immune Response	Reference
** *Mycoplasma hyopneumoniae* **	P97 protein	IL-8 secretion	Th1/Th2 immune response	[152]
** *Salmonella strains* **	Flagellin	NF-kB and MAPKs activation. Induction of proinflammatory mediators resulting in the upregulated expression of cytokines, such as TNF-α, IL-6, IL-8, and proinflammatory free radical synthesizing enzymes, such as the inducible nitric oxide synthase	Efficient immune response in elderly subjects immunized with recombinant hemagglutinin influenza–flagellin fusion vaccine (VAX125)	[128,137,141,153]
**Pharmacologically optimized derivate from *Salmonella flagellin***	Recombinant protein-based on Flagellin, Entolimod (CBLB502)	NF-κB-, AP-1- and STAT3-driven immunomodulatory signaling pathway. Induction of CXCL9 and -10	NK cell-dependent activation of dendritic cells followed by stimulation of a CD8+ T-cell	[151,154,155]

## Data Availability

Not applicable.

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
