# Peer review of "Protein-Based Adjuvants for Vaccines as Immunomodulators of the Innate and Adaptive Immune Response: Current Knowledge, Challenges, and Future Opportunities"

_pharmaceutics, 2022, doi:10.3390/pharmaceutics14081671_

Round 1
Reviewer 1 Report
This is an intensive review covering a wide range of studies on protein-based adjuvants. This review summarized protein-based adjuvants (PBAs) obtained from different organisms, including bacteria, mollusks, plants, and humans, presented the PBA structural features and proposed mechanisms of action, and discussed basic preclinical studies and phase clinical trials. In addition, the authors discussed the current challenges to vaccine PBA development, covering recombinant PBAs, pharmaceutical and regulatory considerations, clinical safety concerns, and potential opportunities for use in new-generation human vaccines. This review article was well written.
I suggest a minor revision before publication. Some comments: All the figures should be enlarged to make them clearly presented. Several statements need to be backed up with references (Page 15 Line 565-566, Page 20 Lines 769-771, page 21 Lines 803-805). The sentence on Page 14 Line 538 should be formatted. The sentence on Page 20 Lines 767-768 should be revised.
Reviewer 2 Report
In the review by Diaz-Dinamarca et al the authors have extensively discussed the role of microbial proteins and other natural substances as existing and potential future vaccine adjuvants also for human use. The review concentrates on describing the Toll-like receptor system related to protein-recognizing PRRs. The topic is important and developing relatively fast and thus an interesting review is justified. There are several aspects that can be taken into account in further processing of the manuscript.
1. The title is somewhat vague, since most of the review is concentrating on potential bacterial and other biological substance based adjuvants that interact with the Toll-like receptor system. Thus, the title could be modified to more specifically describe the content of the review.
2. As a whole the review is very long and in places very detailed (description of TLRs and some microbial proteins) and in other places the text is quite vague like several chapters in the later parts of the review. The style is not in the perfect balance.
3. In the early parts of the review the authors could briefly describe how immunity is developed and how humoral and cell-mediated immunity is developed. It could also be mentioned that PRRs include Toll-like receptors, C-type lectin receptors, NOD-like receptors and RIG-I-like receptors. They also have different stimulatory molecules. Just brief description. The authors have concentrated on describing TLR1/2/6, TLR4 and TLR5, which are recognize protein structures. This is perfectly fine.
4. The description of TLR1/2/6, 4 and 5 ligands and mechanisms is quite long. In case the authors decide to shorten the text, these chapters could be shorted to some extent.
5. I do not fully agree that mucosal vaccines would be better than systemic (intramuscular or intradermal) vaccines. It really depends on the disease to be protected and type of vaccine used. It may be that a mucosal vaccine cannot be developed for several diseases and thus in this respect a systemic vaccine is an option.
6. Cancer vaccines pop-up quite suddenly. I am not sure whether cancer immunology and vaccines is a good idea to include in the present review. Perhaps a more focused review could be obtained by concentrating to infectious diseases.
7. The latter part of the review (starting from point 4) could follow the same order in terms of TLRs. Now flagellin is the first ligand to be described. Why not try to follow the same order as in the TLR chapters i.e. TLR1/2/6 ligands, TLR4 followed by TLR5.
Reviewer 3 Report
The review by Díaz-Dinamarca et al. gives a very extensive overview on the topic of protein-based vaccine adjuvants, covering everything from the different types of adjuvants to the potential risks of their use in vaccine formulations.
The paper is well written and provides a comprehensive insight into the topic. However, going over 33 pages, the review touches on too many subtopics, and, in some instances, the details described are too extensive and distract from the main message. In my opinion the paper would greatly improve if some information that is not directly related to the main message would be omitted.
Lines I would definitely remove:
62 – 70 Aluminum salts ….. immunity is lacking.
95 – 101 The emphasis …. [HGMB1].
128 – 129 including …. and flagellin,
360 – 361 Notably, … separately.
Major concerns:
The introduction gives a general overview on adjuvant usage in vaccines. It only talks about vaccines against infectious diseases. Later the section on clinical trials also talks extensively about cancer immunotherapy. It would be good to at least mention this application also in the introduction.
Figure 1 is about TLR signalling. The figure legend contains a list of organisms from which PBAs are derived. This is an important overview that I would not expect in a figure legend, but rather in the introductory chapter on PBA (chapter 2).
Chapter 2.1: at the end of the first paragraph, a short list of the types of proteins that are going to be described would be helpful or a reference to table 1.
Lines 183 – 193 contains a detailed description of the structure of porins. I do not see the relevance of all this detail, since in the paragraphs describing the interaction with TLR2 no mention is made which structure it is that interacts with the receptor or even whether it is the monomers or trimers or higher molecular structures that are used as adjuvants.
The paragraph about BCG is very confusing, especially as BCG does not contain ESAT-6.
The paragraph on S. pneumonia hsp clearly states they are TLR4 not TLR2 ligands. Why mention them in this chapter? It would make more sense to just have them in TLR4 chapter and mention there that was one paper that showed that TLR2 was somehow involved in reactivity although not via TLR2-signalling.
Chapter 2.2: “TLR2 and ganglioside GM1-dependent activation by PBAs”: since your examples bind to both GM1 and GD1a, this should be changed to “TLR2 and ganglioside-dependent …”.
Lines 315 – 323 are about a TLR2 ligand, this paragraph should be moved to chapter 2.1.
Chapter 2.3: this chapter is especially long and longwinded, try to find ways to shorten this section or giving it a clearer structure. It would again be good to have in the beginning an overview of the proteins that are going to be described or a clear reference to table 2.
Line 434: would be good to first give a list of the proteins from M. tuberculosis before giving the details
Lines 366 – 379 are just confusing.
Chapter 2.4: There is a very detailed description of the structure of flagellin, I again fail to see the relevance of all the minute details, especially as is then not explained which subunits are used in the adjuvants described, f.ex. CBLC502.
Chapter 4.1: Most of the first paragraph describes pre-clinical data and would fit much better into chapter 2.4. On the other hand, chapter 2.4 contains information about clinical studies on Entolimod, which would fit better into this chapter. It would be helpful to call flagellin “TLR5 ligand flagellin” at the first mentioning.
Chapter 4.2: It would be good to mention somewhere that this chapter is about the TLR2-ganglioside ligands.
Chapter 4.3: “information on 151 clinical trials registered with the NIH is currently available”. Why did you choose to elaborate on the studies with STn and GD3 specifically?
Please order the information, first finish all the trials with KLH, and then include the paragraph lines 851 – 859, and then the paragraph 881 – 893.
Paragraph 902 – 909 contains general information, not clinical study data, move or remove.
Chapter 5 and Chapter 5.1: most of the information given is not specific for PBAs. Please shorten and only give the relevant information why it is necessary to produce PBAs in recombinant form, what has been done and what are the specific challenges.
Chapter 6.2: I do not see the relevance of the information given. In contrast to the rest of the paper, which contains too much detail, this seems truncated and short and fails to give context.
Chapter 7.1: Line 1107, this is the first time that two different formulations of KLH are mentioned.
Minor concerns
Line 69: safe and effective vaccine adjuvants: seems to imply the ones currently in use are not safe and effective
Line 72: the “traditional” development….: what does traditional mean in this context?
Line 91 and others: “phase clinical trials”: what does this mean, a summary of phase I – III clinical trials or just a synonym for clinical trials in general?
Line 95: classical and next-generation PBAs: what does that signify?
Line 202: remove “on the other hand”
Line 294: “the cholera B subunit”: should this not be the whole CT?
Line 315: BP1596 should be BP1569
Figure 2: would be good to indicate the location of the subunits of CT, which one binds where
Table 2: PBA CT does not come from species Vibrio Cholerae and E. coli
Figure 5 figure legend: “PBAs can interact with T cells”: actually they do not, they interact with the APC that in term interacts with the PBA, as it is also shown in the figure

Author Response
Please see the attachmnt.

Round 2
Author Response
Thank you again for your careful review of our manuscript and valuable comments. We appreciate your suggestion and have changed the word "promissory" to "encouraging".